# Development and Verification of Infrastructure-Assisted Automated Driving Functions

Martin Rudigier , Georg Nestlinger , Kailin Tong and Selim Solmaz *

Virtual Vehicle Research GmbH, Inffeldgasse 21a, 8010 Graz, Austria; martin.rudigier@v2c2.at (M.R.); georg.nestlinger@v2c2.at (G.N.); kailin.tong@v2c2.at (K.T.)
* Correspondence: selim.solmaz@v2c2.at; Tel.: +43-316-873-9730

**Abstract:** Automated vehicles we have on public roads today are capable of up to SAE Level-3 conditional autonomy according to the SAE J3016 Standard taxonomy, where the driver is the main responsible for the driving safety. All the decision-making processes of the system depend on computations performed on the ego vehicle and utilizing only on-board sensor information, mimicking the perception of a human driver. It can be conjectured that for higher levels of autonomy, on-board sensor information will not be sufficient alone. Infrastructure assistance will, therefore, be necessary to ensure the partial or full responsibility of the driving safety. With higher penetration rates of automated vehicles however, new problems will arise. It is expected that automated driving and particularly automated vehicle platoons will lead to more road damage in the form of rutting. Inspired by this, the EU project ESRIUM investigates infrastructure assisted routing recommendations utilizing C-ITS communications. In this respect, specially designed ADAS functions are being developed with capabilities to adapt their behavior according to specific routing recommendations. Automated vehicles equipped with such ADAS functions will be able to reduce road damage. The current paper presents the specific use cases, as well as the developed C-ITS assisted ADAS functions together with their verification results utilizing a simulation framework.

**Keywords:** automated vehicles; ADAS/AD functions; C-ITS; IVIM; infrastructure assistance; routing recommendations

## 1. Introduction

Advanced driver assistance systems (ADAS) are becoming more widespread in modern automotive vehicles with the promise to reduce the cognitive load on drivers and to increase the driving safety in general. Such assistance systems are also helping with the user acceptance of automated driving (AD) systems with the view of a fully autonomous future mobility. In doing so, however, there are still many hurdles to be cleared in reaching this dream, especially in controlling and effecting the automated vehicle behavior in dynamic traffic conditions. This is particularly important for the transition phase from conventional to automated traffic at different penetration rates.

Automated vehicles that we have on public roads today are capable of up to SAE Level-3 autonomy, which implies conditional automation according to the SAE J3016 Standard taxonomy of automated road vehicles [1]. At this level of autonomy, the driver holds the main responsibility for ensuring the driving safety. Although, all the decision-making processes of this class of automated vehicles depend on the computations performed on the ego vehicle, utilizing only on-board sensor information. This procedure, mimics the perception process of a human driver. However, it is conjectured that for higher levels of autonomy (SAE Level-4 and Level-5), infrastructure assistance will indeed be helpful to ensure partial or full responsibility of the driving safety. In the simplest terms, infrastructure communication can be utilized to convey the dynamic traffic and hazard information ahead and beyond the range of on-board sensors in the form of real-time routing and driving recommendations for connected automate vehicles (CAV).

Motivated by this, the EU-H2020 funded project ESRIUM [2] strives for the high-level goal of increased safety and resource efficiency of transport on European roads. In doing this, its key innovation is a digital map of road surface damage and road wear. This shall help reduce both the number of road works and the associated problems by using new digital services for managing the traffic, as well as by controlling the utilization of the road. The road wear map will contain unique information for the road operators to enhance the road maintenance planning, as well as to provide route and driving recommendations to CAVs and connected vehicles. These recommendations will be used for adjusting the driving path (in-lane and between lanes), which shall help with gradual degradation of the road surface, thereby reducing the regular maintenance actions.

The development of CAVs and infrastructure-assisted automated driving functions have attracted increasing attention in recent years, as summarized in [3,4]. Particularly, lane change and merge maneuvers present a challenge for vehicle automation. The authors in [5] provided a survey in this area, including vehicle positioning systems, communication systems, control systems and system validation. The energy saving potentials of CAVs, on the other hand, have been highlighted in [6]. The conservative assessment in [6] shows 3% energy saving in highway driving in presence of static environment information, and 10% energy saving in arterial driving if traffic signals are provided via vehicle-to-infrastructure (V2I) communications. However, the effect of automated vehicles (AVs) on mixed traffic might be negative. The authors in [7] used real traffic data together with the traffic simulation tool simulation of urban mobility (SUMO) [8] to accomplish the traffic efficiency analysis of mixed traffic, and they claimed that maximum traffic flow rate reduces with higher penetration rate of AVs in mixed traffic. Another important application for CAVs is truck platooning and many problems remain unsolved [9].

In addition to perception and control, trajectory planning is an indispensable function to realize high level of autonomous driving [10]. The configuration space of AVs or CAVs on a 2D plane comprises three dimensions: two dimensions indicating the vehicle's position $(x, y)$, and one dimension indicating the vehicle's heading $\theta$. However, the time dimension or velocity dimension has to be added, if differential constraints and vehicle dynamics are considered. Quite a lot of work tries to tackle this high dimensional problem using the so-called spatiotemporal planning. McNaughton and et al. proposed conformal spatiotemporal lattice to represent search space for structured environments [11]. However, the high dimension of the state space makes the lattice expensive to be repeatedly constructed and searched in a dynamical environment. Recently, Sun and et al. attempted to address this problem by using intelligent driver model (IDM) as a velocity feedback policy [12]. Therefore, the number of dimensions of the lattice planner is reduced.

To balance the trade-off between computation time and quality of the trajectory planning result, a framework of combining graph search and optimization was proposed in [13] and has been augmented recently. In [14,15], an A* search is used to find a rough reference spatiotemporal trajectory along with a collision-free driving corridor, which is subsequently refined by optimization considering safety and dynamical constraints. Moreover, [16] proposed a different approach to identify the spatiotemporal constraints (collision-free driving corridors) using set-based reachability analysis. By combining their approach with optimization-based spatiotemporal planning, arbitrary traffic scenarios can be solved.

Wheel wander, which is related to road surface damage, is of interest for infrastructure designers. Gungor analyzed the literature about wheel wander and its impact on the pavement [17]. According to this report, wheel wander was defined as uncertainty of the lateral position of wheel loads on a lane. The lateral position of the wheels of human driven cars is not uniformly distributed over the whole lane but can be modeled as a normal distribution with non-zero mean and known standard deviation. It was shown in simulation, that wheel wander reduces rutting in comparison to the case without wheel wander.

CAVs, especially heavy-duty trucks in platoon formations, are per design expected to have no wheel wander, and without any measure will lead to more road damage in the from of rutting of the road surface. This is, in fact, very different than the effects of human driven trucks on the road networks, and can cause infrastructure maintenance issues as the penetration of such vehicles increase. In the recent paper [18], this problem was also described and some counter measures were analyzed. Particularly, optimization strategies using V2I communication are proposed for a desired lane off-set the vehicles should drive.

On another strand of work, Bouchihati [19] analyzed the impact of truck platooning on pavement structure on Dutch motorways. It is stated in this publication that automated truck platoons will lead to substantial higher damage on the pavement as result of rut development and fatigue. As a potential solution, smart lanes were proposed. Accordingly, sensors can be used to measure the positions of all previous passed truck platoons and the optimum lateral position can be communicated to the leading truck who will then adjust its lateral position according to the given recommendation. With this method, an optimal use of the road surface can be achieved.

Inspired by these, the EU project ESRIUM investigates infrastructure assisted routing and driving recommendations utilizing C-ITS communications. In this respect, specially designed ADAS functions (i.e., lateral and longitudinal tracking controllers and a trajectory planner) are being developed with capabilities to adapt their behavior according to specific routing and driving recommendations received in the form of infrastructure to vehicle information message (IVIM). The current paper presents the specific use-cases to demonstrate the utilization of this approach for reduction in road rutting (due to reduced wheel wander of CAVs), and to initiate informed lane changes to avoid potential hazard situations. The paper further describes the developed C-ITS assisted ADAS functions together with their verification results utilizing a simulation framework.

The paper is organized as follows: First, we introduce the two use cases of the C-ITS assisted routing and driving recommendation in Section 2. In Section 3, the simulation framework is introduced. Next, the ADAS functions are defined in Section 4. Simulation implementation and verification results are finally given in Section 5 followed by the conclusions and outlook in Section 6.

## 2. Descriptions of the Use Case Scenario and Performance Indicators

In this paper, two use case scenarios of ESRIUM are described. The first one is an in-lane off-set recommendation for connected automated vehicles (CAVs) and the second is a strategic lane change and lane utilization information for CAVs. In both cases, the ego vehicle receives the information via the C-ITS message type IVIM [20].

In the in-lane off-set recommendation scenario, as sketched in Figure 1, the ego vehicle shall drive in automated mode (SAE Level-3 equivalent Motorway Chauffeur combining adaptive cruise control (ACC) and lane keeping assist (LKA) driving functions) in a detection zone when it receives an IVIM containing a recommended lane-off-set information. This C-ITS routing recommendation in the form of a IVIM comes from an infrastructure road-side unit (RSU) and is received by an on-board unit (OBU) and the linked automated driving functions on the ego vehicle interpret it. The detection zone defines the region, where the vehicle is expected to be in a bi-lateral communication link with the RSU. Before entering the relevance zone (this is the zone, where the recommended action by the IVIM needs to be implemented), the ego vehicle adapts the typical LKA task of tracking the center of the existing lane and transitions to driving along the same lane with the given in-lane off-set. The ego vehicle is expected to drive throughout the relevance zone with the recommended in-lane off-set in case traffic conditions permit it. Immediately after leaving the relevance zone, the ego vehicle is expected to follow the default centerline tracking task unless otherwise recommended.

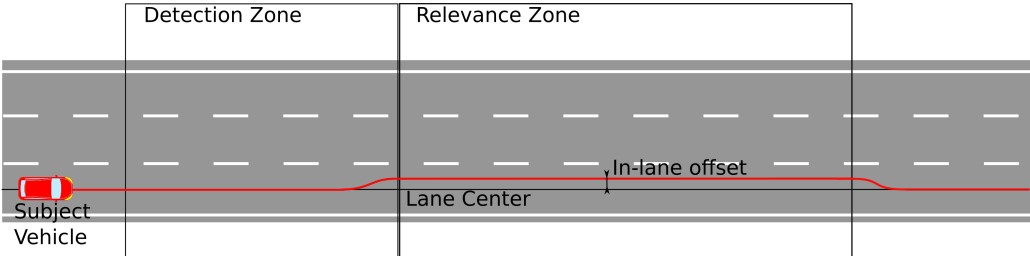

**Figure 1.** In-lane off-set recommendation scenario description.

In the lane change recommendation scenario (see Figure 2) , the ego vehicle shall drive in automated mode (SAE Level-3 equivalent Motorway Chauffeur combining ACC and LKA driving functions) in a detection zone when it receives an IVIM containing a set of three relevance zones with instructions to avoid the rightmost lane, because of road damage. When the ego vehicle is driving on the rightmost lane in relevance zone 1, it will change the lane to next lane if traffic allows it, and in relevance zone 2 it will continue driving on the remaining lanes and avoid the rightmost lane. In relevance zone 3, the rightmost lane is cleared and the ego vehicle uses the most appropriate lane according to the traffic situation.

**Figure 2.** Lane change recommendation scenario description (see Table 1 for the definitions of the pictogram symbols)

## 3. Simulation Environment and Setup

### 3.1. Driving Functions and Vehicle Dynamics

For the development of the driving functions, a co-simulation environment based on Matlab/Simulink and the vehicle dynamics software IPG Carmaker was utilized (see Figure 3a). This simulation environment was preferred since it is well suited for rapid-prototyping purposes. As depicted in the figure, all driving functions in terms of trajectory planner, as well as longitudinal and lateral tracking controllers were developed and implemented in Matlab/Simulink. Those include the vehicle state, an object list and lane marking information. The vehicle state includes actuation signals (steering angle, brake and gas pedal), velocities and accelerations. For the object list, no sensor dynamics were assumed considering only occlusion effects. Taking into account limitations of a prospective real world demonstration of the presented implementations, lateral vehicle guidance was based on lane marking information only. For that purpose, a polynomial lane marking model was recreated in simulation, which describes lane boundaries as third order polynomials and related polynomial domains.

The routing recommendations based on IVIM messages were implemented as preset messages. Future demonstrations will be performed on sections of the Austrian motorway A2. Therefore, the corresponding UHDmaps(®) [21] in the OpenDRIVE format was used in the CarMaker driving scenario.

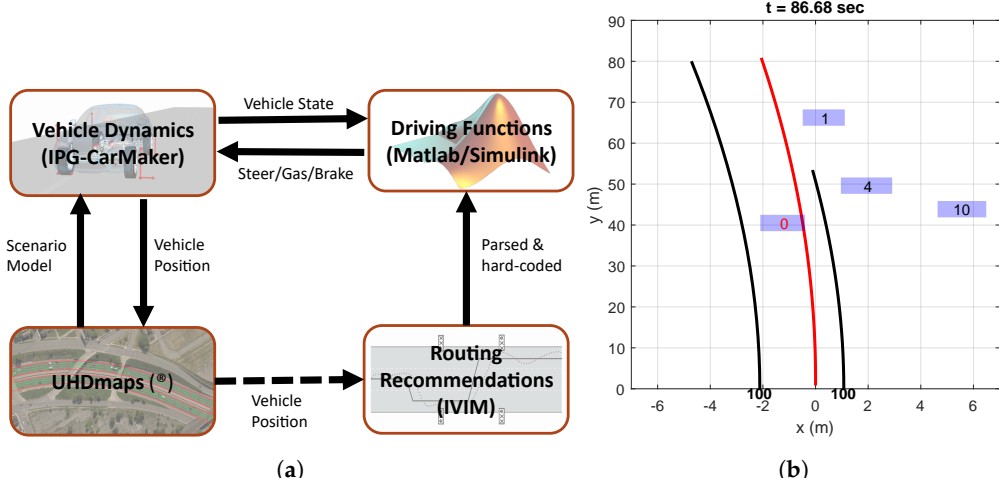

**Figure 3.** Simulation environment, setup, and ego vehicle aligned visualization. (**a**) Co-simulation architecture utilizing Matlab/Simulink and IPG Carmaker. (**b**) Visualization of objects (blue boxes) and assigned ID, lane markings and quality of detection (black), and ego prediction (red).

Object list and lane marking inputs to the driving function were visualized in an ego-centered view online (Figure 3b), i.e., in parallel with the simulation. This allowed to validate driving function inputs and outputs.

### 3.2. C-ITS Message Structure and Simulation Implementation

In the final implementation, the recommendations for lane off-set and lane change will be sent using the IVIM messages. IVIM information will be broadcast according to TS 103 301 V2.1.1 and ISO 19321: 2020 standard. The automated vehicle container of the broadcast IVIM message will include the pictograms of the recommended action. In the case of lane change recommendations, one of the four categories of recommendations associated with corresponding pictograms according to the ISO 14823:2017 "Intelligent transport systems-Graphic data dictionary", as indicated in Table 1, will be used.

**Table 1.** IVIM lane choice recommendation according to the ISO 14823:2017 code.

| Recommendation | ISO14823 Code | Pictogram |
|---|---|---|
| Keep the current lane | 13,660 | ↓ |
| Move to the left lane | 13,661 | ↙ |
| Move to the right lane | 13,662 | ↘ |
| Lane closed | 13,669 | ✗ |

The lane off-set recommendation will also be broadcast using the free text option in the automated vehicle container of the IVIM message. The free text will contain the signage followed by the off-set in centimeters. In this setting "+" sign indicates an off-set to the right of the lane center whereas "−" sign indicates an off-set to the left relative to the lane center. The example information content of the lane off-set IVIM message is given in Table 2.

**Table 2.** IVIM lane off-set messages information structure.

| AV-Container Part | Detection-Zone Ids | Relevance-Zone Ids | Applicable Lane | ±Offset [cm] |
|:---:|:---:|:---:|:---:|:---:|
| 1 | 1 | 11 | 1 | −20 |
| 2 | 1 | 13 | 2 | 10 |
| ⋮ | ⋮ | ⋮ | ⋮ | ⋮ |

In the scope of the simulation based development of the driving functions, the implementation of sending and receiving of the IVIM communications is not considered. Although parsing of the message standard structure described above is not part of this first simulations. For both use case scenarios a simple emulation subsystem was developed. Its aim is to check if the ego vehicle is inside the detection or relevance zones and to provide the trajectory planner with the necessary information. In the case of the first scenario (i.e., in-lane off-set recommendation), this information is the desired off-set for each lane in a give relevance zone. For the lane change recommendation scenario, on the other hand, the specific information that is provided to the driving functions includes one detection zone and three relevance zones. Therefore, the developed emulation block checks and interprets whether the ego vehicle is inside one of these areas and consequently specifies the desired lane to the trajectory planner. The specific implementation of the IVIM parser will be conducted as part of the real vehicle tests. The ego vehicle has to be equipped with a GNSS-receiver and compares its position with the definitions of the detection and relevance zones to evaluate if it is inside of a zone.

## 4. AD Functions Development

The AD functions presented in this article are discussed in the well known sense–plan–act scheme. As mentioned in Section 3, the sensing task is delegated to CarMaker, which provides all the required ego vehicle states, as well as obstacle and road information. The sensors are therefore modeled as low-fidelity ideal sensors providing object lists within specified cones representing their respective field of view. The planning task is accomplished by a rule-based trajectory planner adapted to utilize C-ITS messages, as described below in Section 4.1.

### 4.1. Planning: Rule-Based Trajectory Planner

The planning task is accomplished by a rule-based trajectory planner (TP) that was developed for structured environments like highways with well defined lane boundaries. It uses a finite state machine and a set of discrete decisions to trigger lane changes or keep the vehicle on its current lane. By default, the ego vehicle drives in the middle of the rightmost lane. If a slower vehicle prevents the ego vehicle from reaching its desired cruising speed, and the target lane is not occupied, a Bézier curve is planned to perform a lane change.

The value of the Bézier curve at a tunable look-ahead distance is then used as the desired lateral off-set for the underlying lateral controller. As can be seen from Figure 4, this off-set is defined with respect to the center of the current lane. Therefore, the TP needs to handle a change of reference during the lane change maneuver. This allows future deployment of the proposed algorithm to a demonstrator vehicle which utilizes only a vision sensor for lane marking based in-lane localization.

Together with emulations blocks for parsing the IVIM (see Section 3.2), the TP is able to generate the input signals for the lateral and longitudinal controllers to accomplish the described scenarios in Section 2.

Bézier Based Path Planning Approach

Path planning relies on a Bézier curve approach [22] that was enhanced to handle specific limitations that are common in real world applications. Apart from using HD

maps, currently available sensors lack the ability of providing trustworthy lane information, specifically the lane width, besides the current lane. To overcome this issue, the planned path consists of two quadratic Bézier curves connected via a straight segment as shown in Figure 4.

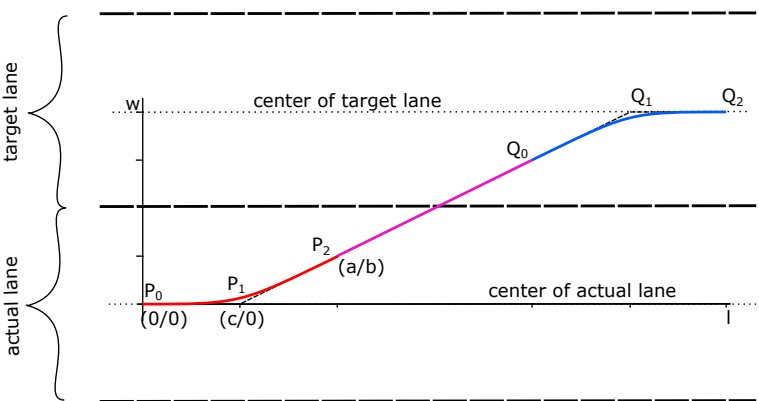

**Figure 4.** Lane change with two quadratic Bézier curves and a linear segment.

The Bézier curves provide a smooth transition to initiate and finish the lane change, while the straight segment can be shortened or extended during the lane change maneuver according to the width of the target lane.

The path as shown in Figure 4 can then be written as a spline

$$S(\tau) = \begin{cases} \Gamma_1(\tau) & 0 \leq \tau \leq 1, \\ \Gamma_2(\tau) & 1 \leq \tau \leq 2, \\ \Gamma_3(\tau) & 2 \leq \tau \leq 3. \end{cases} \tag{1}$$

with three segments $\Gamma_i$ ($i = 0, 1, 2$) and the path parameter $\tau$. Considering a quadratic Bézier curve

$$\mathcal{C}(\tau, C_0, C_1, C_2) = C_0(1 - \tau)^2 + 2C_1(1 - \tau)\tau + C_2\tau^2, \quad \tau \in [0, 1] \tag{2}$$

with control points $C_i$ ($i = 0, 1, 2$), the spline segments $\Gamma_1$ and $\Gamma_3$ were chosen as

$$\Gamma_1(\tau) := \mathcal{C}(\tau, P_0, P_1, P_2) \quad \text{and} \quad \Gamma_3(\tau) := \mathcal{C}(\tau, Q_0, Q_1, Q_2). \tag{3}$$

For the terminal points of $S(\tau)$, we can immediately state from Figure 4:

$$P_0 = (0, 0) \quad \text{and} \quad Q_2 = (l, w). \tag{4}$$

To achieve a comfortable lane change, the length $l$ of the lane change is chosen according to the current vehicle speed, while the lane change off-set $w$ is pre-defined by the widths of the current and target lane:

$$w = \frac{w_{\text{actual}} + w_{\text{target}}}{2}. \tag{5}$$

Since $\Gamma_3$ is skew-symmetric to $\Gamma_1$, the relations

$$Q_0 = Q_2 - P_2 \quad \text{and} \quad Q_1 = Q_2 - P_1 \tag{6}$$

hold, leaving only $P_1 := (c, 0)$ and $P_2 := (a, b)$ undetermined. The straight segment $\Gamma_2$ shown in magenta connects the points $P_2$ and $Q_0$, therefore

$$\Gamma_2(\tau) = (Q_0 - P_2)\tau + P_2 = (Q_2 - 2P_2)\tau + P_2. \tag{7}$$

Claiming geometric continuity at the transition from $\Gamma_1$ to $\Gamma_2$ (and, therefore, from $\Gamma_2$ to $\Gamma_3$), the relation

$$\frac{b}{a-c} = \frac{w-2b}{l-2a}. \tag{8}$$

must be fulfilled. According to Figure 4, it is reasonable to restrict $b \leq \frac{1}{2}w_{\text{actual}}$, which is usually fulfilled by setting $b$ to half the width of the ego vehicle. Then, introducing $f_c = \frac{c}{a}$ as a tunable parameter, (8) can be solved for

$$a = \frac{bl}{w + 2bf_c - wf_c}. \tag{9}$$

To conclude the path planning approach, the control points are

$$\begin{aligned} P_0 &= (0,0), \quad P_1 = (c,0), \quad P_2 = (a,b), \\ Q_0 &= (l-a, w-b), \quad Q_1 = (l-a/2, w), \quad Q_2 = (l,w). \end{aligned} \tag{10}$$

As mentioned, $b$ was set to half the ego vehicle width, while $f_c$ was determined empirically considering the trade-off between the slope and continuity of the spline $S$. For the simulations presented in Section 5, $b = 0.9$ and $f_c = 0.5$ were used.

### 4.2. Actuation: Lateral and Longitudinal Control

For tracking the lateral and longitudinal references from the trajectory planner, two proven controller implementations were reused. The lateral controller implements a state-feedback control law based on [23], modified according to [24] as motivated from a demonstrator vehicle application. It makes use of the reference path from the trajectory planner up to the second geometric continuity, i.e., lateral error, heading error, and path curvature. The actuated signal is the steering wheel angle. Although, the dynamics of a steer-by-wire actuator are considered.

The longitudinal controller is implemented as a time-discrete PI (proportional-integral) controller with an anti-wind-up measure providing the control signal $p_k$ at time step $k$ according to

$$p_k = k_{\text{P}} e_k + \alpha_k \tag{11}$$

$$\alpha_k = k_{\text{I}} e_k + \max(-100, \min(\alpha_{k-1}, 100)). \tag{12}$$

Here, $e_k$ is the acceleration error, $k_{\text{P}}$ and $k_{\text{I}}$ are the proportional and integral gain and $p_k$ is the commanded brake pedal position for $p_k < 0$ and the throttle position for $p_k > 0$. The anti-wind-up measure is implemented via $\max()$ and $\min()$ functions ensuring $p_k \in [-100, 100]$. To ensure bumpless activation of the longitudinal controller, $\alpha_0$ is initialized according to the current pedal positions.

Both controllers execute with a sample time of 20 ms.

## 5. Simulation Results and Analysis

The described use case scenarios in Section 2 were simulated on a straight section of the A2 motorway without other traffic elements. The vehicle speed was set to 130 km/h for the longitudinal tracking controller.

Figure 5 shows the results of the in-lane off-set recommendation scenario. For this simulation a detection zone with a length of 200 m was chosen and a relevance zone with a length of 1000 m and the in-lane off-set recommendation is only valid for the rightmost lane. In Figure 5a, the steering wheel angle during the maneuver is depicted. Moreover, the lateral acceleration in Figure 5b, the yaw rate in Figure 5c, and the desired and driven off-sets in Figure 5d are shown, respectively.

According to the simulated scenario, at approximately 2.5 s the ego vehicle starts the transition to the desired in-lane off-set. In this example the vehicle reaches its desired steady-state off-set value of $-0.2$ m at approximately 7 s. The total time it took to reach the

set off-set value is, therefore, about 4.5 s, which matches the in the controller defined value. According to the designed scenario, the ego vehicle drives through the relevance zone (RZ) with the desired off-set and leaves it at about second 35 returning back to zero off-set value.

As seen from the results in Figure 5, the developed enhanced driving functions are quite effective in achieving the desired off-set value. The difference between the desired and achieved off-set values are quite small except a small overshoot. The overshoot in the lateral position was smaller than 2 cm in this case. The small oscillation in the middle point of the maneuver (which should not be existing at all, at about 21 s) was caused by the CarMaker itself and is believed to be a modeling bug, as it varied when the vehicle model was changed, but could not be avoided completely.

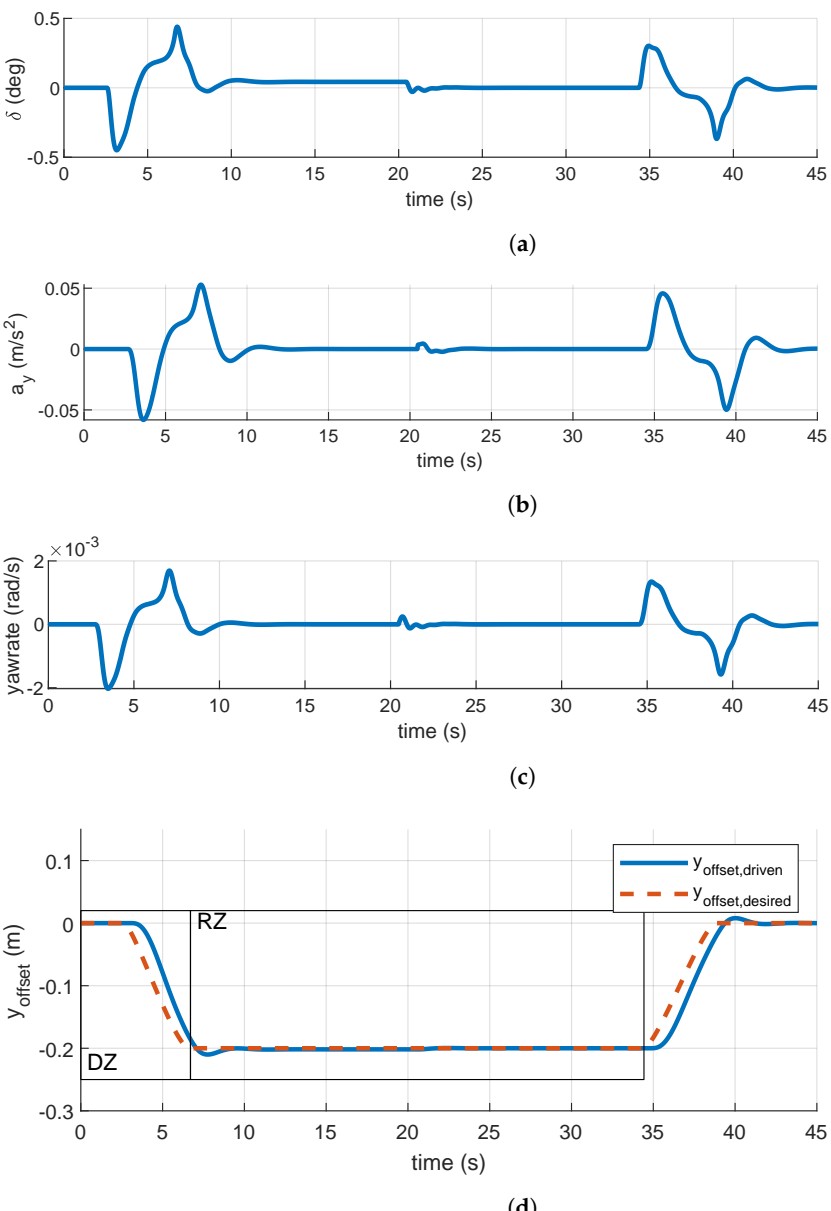

**Figure 5.** Result of the in-lane off-set recommendation scenario: (**a**) steering wheel angle, (**b**) lateral acceleration, (**c**) yaw rate, and (**d**) lane off-set.

Figure 6 shows the use case scenario simulation results for the strategic lane change and lane utilization recommendation for CAVs. Figure 6a shows the steering wheel angle variation during this example recommended lane change maneuver. Furthermore, the lateral acceleration in Figure 6b, the yaw rate in Figure 6c, and, finally, the vehicle lateral

position relative to the desired lane in Figure 6d are shown, respectively. According to this example scenario, the ego vehicle enters the relevance zone 1 (RZ1) at about 6.6 s and the desired lane, therefore, switches from 0 to 1. Note that in the TP the rightmost lane is indexed with 0 and the index is increased while going towards the left side, which is different to the numbering convention in the IVIM. Since there is no hindering traffic in the scenerio, the TP initiates immediately a lane change maneuver to lane 1, where it stays while passing the relevance zones 1 and 2 (RZ1 and RZ2). In relevance zone 3 (RZ3) the TP computes a lane change maneuver back to the original starting lane (i.e., lane 0). In this use case as well, the recommended maneuvers were conducted effectively with little overshoot. It can be observed that the total lane change maneuver was achieved with less than 5 s transition times between steady state driving positions.

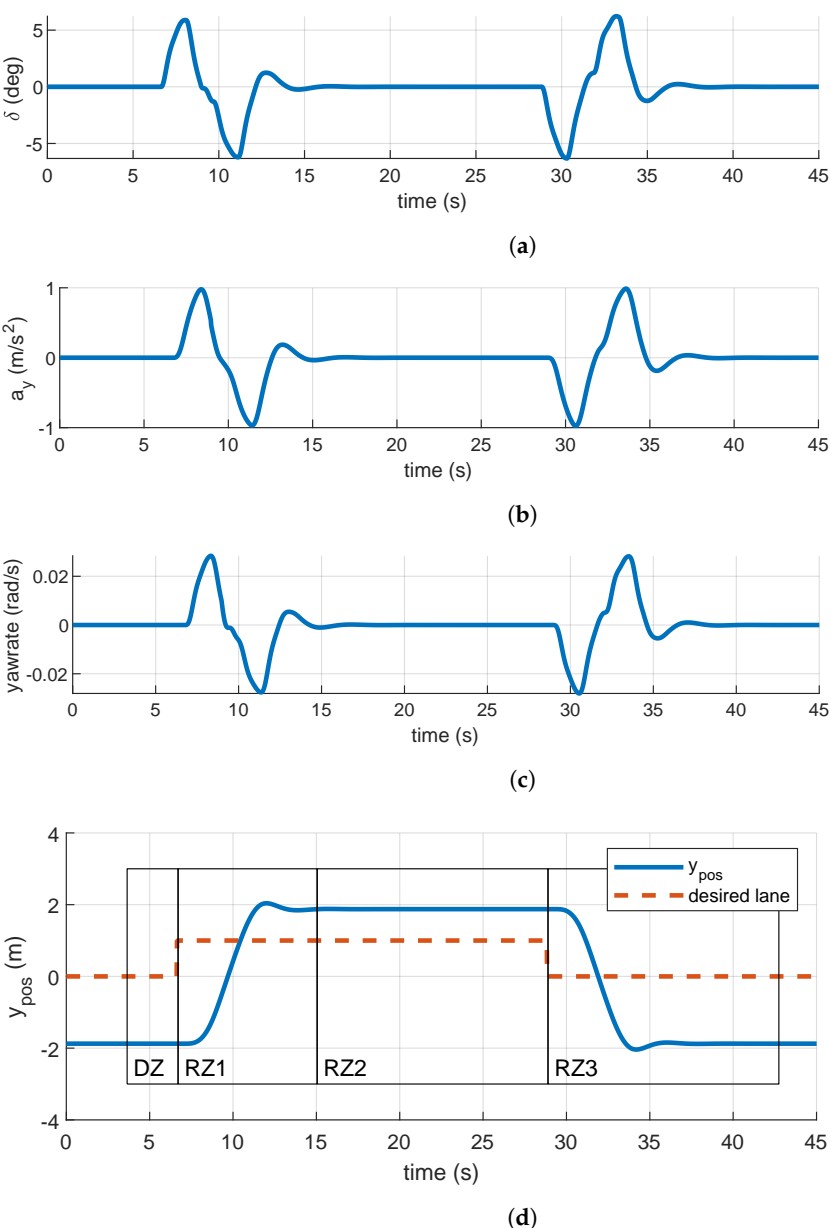

**Figure 6.** Result of the lane change recommendation scenario: (**a**) steering wheel angle, (**b**) lateral acceleration, (**c**) yaw rate, (**d**) lateral vehicle position.

As a further simulation use case example, we implemented also a combined in-lane off-set and lane change recommendations, as seen in Figure 7. In this scenario, the ego vehicle starts in the right lane and first receives a in-lane off-set recommendation and than

consecutively a lane change to the left lane followed by reverse maneuvers to return to the right lane lateral 0-off-set position. In this case, the in-lane off-set values were varied with specific values {−1 m, −0.5 m, −0.2 m, −0.1 m, 0 m, 0.1 m, 0.2 m, 0.5 m, 1 m} and the resulting trajectories were displayed in Figure 7 as a superimposed plot. The results indicate that complex maneuvers combining the two specific recommendations can also be achieved with efficacy.

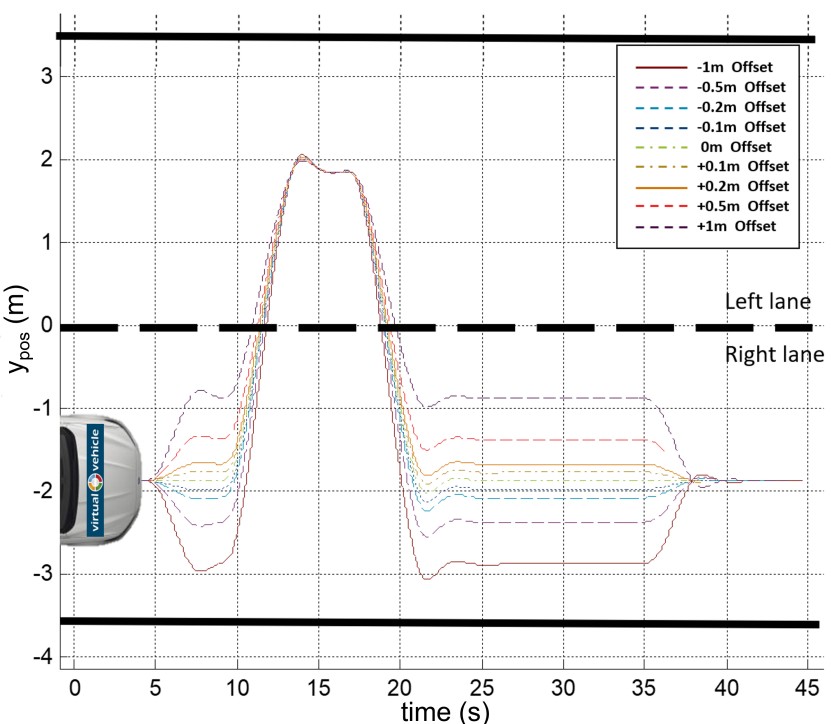

**Figure 7.** Combined in-lane off-set and lane change recommendations with varying lateral off-set values. The lane boundaries were shown to aid the visualization.

## 6. Conclusions and Outlook

Given that the number of automated vehicles increase rapidly, their behavior on the motorways can have abnormal ramifications on the road surface wear. Consider, for example, automated truck platoons driving along same road stretch, which will surely lead to frequent rutting. Motivated by this, we introduced in this paper a new model for operation of automated driving functions, namely by using specific infrastructure routing recommendations for enhancing or adapting their behavior.

The recommendations are in the form of specific lane-change and in-lane off-set suggestions. It is envisaged that from the collective behavior of the automated vehicles (and connected vehicles in general) with a capability to adapt their behavior according to the routing recommendations will benefit both the road operators and the vehicle owners. These vehicles will, on the one hand, reduce rutting of the surface, as well as unbalanced wear of the road surface, especially when their penetration rates increase. On the other hand, such vehicles will avoid rutted or damaged road sections and blocked lanes thanks to the routing recommendations, providing a safer and more comfortable ride. This is an idea being developed within the recently started EU-H2020 funded project ESRIUM, where the complete value chain for implementing this basic idea is being developed to investigate its feasibility.

In this paper we described the specific ADAS/AD functions including longitudinal and lateral tracking controllers, as well as a rule based trajectory planner, which are utilized to achieve the routing recommendations. We also introduced the specific use cases that will be implemented in this context, and the structure of the routing recommendations based on the IVIM message standard. The corresponding ADAS/AD functions were developed in a

high-fidelity simulation framework based on Matlab/Simulink and Carmaker software with the view of deploying the same functions in a test vehicle for real-life implementation. The ADAS/AD functions were demonstrated for representative use cases to verify their basic operation. In the future extensions, the same driving functions will be imported to an automated drive demonstrator vehicle to implement and realize the demonstration of the routing recommendations in real-life use cases on a public road. It is planned to conduct this demonstration during the progress of the ESRIUM project.

**Author Contributions:** Conceptualization, M.R., G.N. and S.S.; methodology, M.R., G.N. and S.S.; software, M.R. and G.N.; validation, M.R., G.N. and S.S.; formal analysis, M.R., G.N. and S.S.; investigation, M.R., K.T. and S.S.; data curation, M.R and G.N.; writing—original draft preparation, M.R., G.N., K.T. and S.S.; writing—review and editing, M.R., G.N., K.T. and S.S.; visualization, M.R., G.N. and S.S.; supervision, S.S.; project administration, S.S.; All authors have read and agreed to the published version of the manuscript.

**Funding:** This work was conducted in the scope of the ESRIUM Project, which has received funding from the European Union Agency for the Space Programme under the European Union's Horizon 2020 research and innovation programme and under grant agreement No 101004181. The content of this paper reflects only the authors' view. Neither the European Commission nor the EUSPA is responsible for any use that may be made of the information it contains.

**Acknowledgments:** Virtual Vehicle Research GmbH has received funding within COMET Competence Centers for Excellent Technologies from the Austrian Federal Ministry for Climate Action, the Austrian Federal Ministry for Digital and Economic Affairs, the Province of Styria (Dept. 12) and the Styrian Business Promotion Agency (SFG). The Austrian Research Promotion Agency (FFG) has been authorised for the programme management.

**Conflicts of Interest:** The authors declare no conflict of interest. The content of this paper reflects only the authors' view. Neither the European Commission nor the EUSPA is responsible for any use that may be made of the information it contains. The funders had no role in design of the study, in the collection, analyses, or interpretation of data; in the writing of the manuscript, or in the decision to publish the results.

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
