# Peer review of "Development and Verification of Infrastructure-Assisted Automated Driving Functions"

_electronics, doi:10.3390/electronics10172161_

Round 1

Reviewer 1 Report

The paper presented an infrastructure-assisted automated driving function, namely a lane changing/offset function based on road surface conditions. A new lane changing/offset has been developed and simulated with simplest traffic condition. The overall novelty and impact to AV/CAV development is modest. Comments to the paper are listed in the attached document. Some major comments are listed as follows:

  1. There are a few grammatical errors and awkward sentences as indicated in the attached comments summary. The authors need to correct them.
  2. The presentation and description of some figures and equations need to be improved by giving more details, as indicated in the attached comment file.
  3. In the description of use cases, the AD function development and the simulation, the authors did not address when the ego vehicle made the decisions that it was entering the different zones, and based on what (e.g. GPS, or other info) information it made the decision. The authors needs to describe this somewhere briefly.
  4. The paper did not compare its results with any other similar work, for example, the efficacy of the TP planning in avoiding wane road or potholes. If the authors are not able to find similar work to compare with or it is not easy to repeat other's work, the author could compare with the reported results of other work. In general, the authors need to explain why no comparison is given in the paper, or add the comparison result in revision.

Author Response

We thank the Reviewer-1 for his detailed and very helpful review of our paper. We first comment on the marked-up copy of the paper and then answer the specific questions and remarks by the reviewer in the open review comments below.

Page: 1

Number: 1 Author: Reviewer

The authors need to provide some references to support this "widely accepted" statement.

Authors answer:

This sentence was reformulated to a more general statement.

Page: 2

Number: 1 Author: Reviewer

Reference is needed.

Authors answer:

A reference was added.

Number: 2 Author: Reviewer

What "planning"? Is it route planning or other kind of planning?

Authors answer:

It was changed to specifically “trajectory planning”.

Number: 3 Author: Reviewer

Suggest to change to "function"

Authors answer:

It was changed to “function”.

Number: 4 Author: Reviewer

Need to be specific about "the solution". Otherwise, readers could be confused about what/which solution you are talking about.

Authors answer:

It is now specifically about the “trajectory planning result”

Number: 5 Author: Reviewer

Suggest to replace "the" with "a".

Authors answer:

We have replaced it.

Page: 3

Number: 1 Author: Reviewer Subject: Cross-Out Date: 08/08/2021 14:21:24

can be deleted.

Authors answer:

It was deleted.

Number: 2 Author: Reviewer Subject: Highlight Date: 08/08/2021 15:16:10

Reference is needed!

Authors answer:

The reference to the defining standard was added.

Number: 3 Author: Reviewer Subject: Highlight Date: 08/08/2021 14:24:46

Need to give the full text terms of these two abbreviations as this is the first time they are used in this paper.

Authors answer:

Full text terms are now given.

Number: 4 Author: Reviewer Subject: Highlight Date: 08/08/2021 14:28:01

Full text is needed.

Authors answer:

Full text terms are now given.

Number: 5 Author: Reviewer Subject: Highlight Date: 08/08/2021 14:28:33

Full text is needed.

Authors answer:

Full text terms are now given.

Page: 4

Number: 1 Author: Reviewer Subject: Highlight Date: 08/08/2021 14:34:04

What will happen if traffic does not allow it changing lane?

Authors answer:

The vehicle will remain on the damaged lane. Note that we talk about small damages of the road which do not prevent the vehicle from driving on the damaged lane.  The main aim of the system is to prevent the damage from getting larger.

Number: 2 Author: Reviewer Subject: Highlight Date: 08/08/2021 14:37:20

Reference is needed.

Authors answer:

Reference to Mobileye 630 was removed and implemented approach was explained briefly.

Page: 5

Number: 1 Author: Reviewer Subject: Highlight Date: 09/08/2021 09:11:09

What's the unit of values in the x- and y-axis in Fig.3b?

Authors answer:

Units are meter and were added to the figure.

Number: 2 Author: Reviewer Subject: Highlight Date: 08/08/2021 14:41:40

You need to describe/explain the meaning of the curves and the coloured numbers in Fig. 3b, either in this paragraph or given the legends in the figure.

Authors answer:

Explanation for the curves was added.

Page: 6

Number: 1 Author: Reviewer Subject: Highlight Date: 08/08/2021 15:20:12

includes

Authors answer:

Done

Number: 2 Author: Reviewer Subject: Cross-Out Date:  08/08/2021 15:22:42

Authors answer:

Done

Page: 7

Number: 1 Author: Reviewer Subject: Highlight Date: 08/08/2021 15:44:20

Where are Ghe in the figure? You need to indicate them in the figure.

Authors answer:

The figure was also updated to be more readable. Unfortunately, we don’t know to what you are referring by “Ghe”.

Number: 2 Author: Reviewer Subject: Highlight Date: 08/08/2021 15:38:26

But there is no Qi in (2). How is Qi selected in the equation?

Authors answer:

Thanks for pointing this out. This section was reformulated to make the definition of the spline segments and the relation with the Bezier curves control points clearer.

Number: 3 Author: Reviewer Subject: Cross-Out Date: 08/08/2021 15:48:12

Authors answer:

Has been removed.

Number: 4 Author: Reviewer Subject: Highlight Date: 08/08/2021 15:50:32

Why are these specific values used? You need to briefly explain the reasons for using the specific values.

Authors answer:

Please note, that this section was extended by the strategy to select those values.

Page: 8

Number: 1 Author: Reviewer Subject: Highlight Date: 08/08/2021 15:53:24

Which black curve? All curves in Figure 4 are in Black. You have to clearly differentiate or indicate the curve you referred to.

Authors answer:

This reference was removed due to reworking the whole section.

Number: 2 Author: Reviewer Subject: Highlight Date: 08/08/2021 15:59:30

You need to revise the sentence so that it makes sense. For example, what equations or which equations do not need to be changed? Do you mean all equations, (1) to (9), or some of them. Be specific please.

"the equations nothing has to be changed" sounds not right. You probably want to say "the equations do not need to be changed" or "there is nothing to be changed in the equations". Please revise the sentence.

Authors answer:

Reformulated due to reworking the whole section.

Number: 3 Author: Reviewer Subject: Highlight Date: 08/08/2021 16:02:49

Suggest to replace it with "with the assumption"

Number: 4 Author: Reviewer Subject: Highlight Date: 09/08/2021 09:15:28

What is PI controller? From the context, I can understand that you are using "Proportional-integral" controller. Please give the full text terms of the abbreviation. Please also give a reference as not all readers are familiar with this.

Authors answer:

Full text added.

Number: 5 Author: Reviewer Subject: Highlight Date: 08/08/2021 16:08:58

Following this, you need to continue to describe the meanings of other symbols, such as pk, kp, alpha_k, ki and the function of max(), in (10) and (11). And what is k?

Authors answer:

Explanation for unmentioned symbols added.

Number: 6 Author: Reviewer Subject: Highlight Date: 08/08/2021 16:12:00

"sample rate" should not be presented in time "ms", it should be in frequency. What you should say here is "sample interval"

Authors answer:

I think sample time is also a very valid term and sounds more familiar to the authors.

Number: 7 Author: Reviewer Subject: Highlight Date: 09/08/2021 09:30:22

You need to give the details of the different zones (i.e. the detection zone, relevance zones), such as their lengths, widths, etc. Another important issue that the authors did not address is when the ego vehicle make the decides that it gets into the different zones, and based on what (e.g. GPS, or other info) information it makes the decision. The authors needs to describe this somewhere briefly.

Authors answer:

Thank you for your remark. The length of the zones and the valid lane were inserted in the text.

The information about the necessary GNSS receive, to compare the vehicle position with the definitions in the IVIM, was added on another place (Page 5, line 183).

Number: 8 Author: Reviewer Subject: Cross-Out Date: 08/08/2021 16:12:04

Authors answer:

Thank you for remark, the second ‘in’ was deleted.

Number: 9 Author: Reviewer Subject: Highlight Date: 08/08/2021 16:13:04

at ts=2.5s

Authors answer:

Thank you for your remark. We added “at” and additionally  “approximately”.  

Number: 10 Author: Reviewer Subject: Highlight Date: 08/08/2021 16:28:42

You need to indicate ts and tf in figure 5.

Authors answer:

We deleted ts and tf, because they were no defined values. The values are only approximated and should indicate where the reader should look in the figure.

Page: 9

Number: 1 Author: Reviewer Subject: Highlight Date: 08/08/2021 16:17:54

a modeling bug

Authors answer:

Thank you very much for this remark. The article “a” was added.

Number: 2 Author: Reviewer Subject: Highlight Date: 08/08/2021 16:29:19

Suggest to change this to "time (s)" for all figures.

Authors answer:

Thank you for your remark. We followed the suggestion and changed the labels.

Page: 10

Number: 1 Author: Reviewer Subject: Highlight Date: 08/08/2021 16:22:55

You don't know whether it is minimal or not. You could safely say "little".

Authors answer:

Thank you for valuable remark, we followed the advice and exchanged “minimal” through “little”.

Number: 2 Author: Reviewer Subject: Cross-Out Date: 08/08/2021 16:23:51

Authors answer:

Thank you for this remark. The second “the” was deleted.

Number: 3 Author: Reviewer Subject: Cross-Out Date: 08/08/2021 16:24:18

Authors answer:

Thank you for this remark. The “then” was deleted.

Number: 4 Author: Reviewer Subject: Highlight Date: 08/08/2021 16:30:14

wrong label.

Authors answer:

Thank you for this comment, we corrected the label.

Number: 5 Author: Reviewer Subject: Highlight Date: 08/08/2021 16:31:16

"Consider, for example, automated ..."

Authors answer:

Thank you for your comment. The missing colons were added.

Page: 11

Number: 1 Author: Reviewer Subject: Highlight Date: 08/08/2021 16:33:53

suggest to change to "envisaged"

Authors answer:

Thank you for this remark. We followed the suggestion and replaced “conjectured” with “envisaged”.

Number: 2 Author: Reviewer Subject: Highlight Date: 08/08/2021 16:36:43

suggest to replace with "this"

Authors answer:

Thank you for this remark. We replaced “current” with “this”

Open Review

English language and style

( ) Extensive editing of English language and style required
( ) Moderate English changes required
(x) English language and style are fine/minor spell check required
( ) I don't feel qualified to judge about the English language and style

Yes

Can be improved

Must be improved

Not applicable

Does the introduction provide sufficient background and include all relevant references?

()

(x)

( )

( )

Is the research design appropriate?

(x)

( )

( )

( )

Are the methods adequately described?

()

(x)

( )

( )

Are the results clearly presented?

()

(x)

( )

( )

Are the conclusions supported by the results?

(x)

( )

( )

( )

Comments and Suggestions for Authors

The paper presented an infrastructure-assisted automated driving function, namely a lane changing/offset function based on road surface conditions. A new lane changing/offset has been developed and simulated with simplest traffic condition. The overall novelty and impact to AV/CAV development is modest. Comments to the paper are listed in the attached document. Some major comments are listed as follows:

  1. There are a few grammatical errors and awkward sentences as indicated in the attached comments summary. The authors need to correct them.
  2. The presentation and description of some figures and equations need to be improved by giving more details, as indicated in the attached comment file.
  3. In the description of use cases, the AD function development and the simulation, the authors did not address when the ego vehicle made the decisions that it was entering the different zones, and based on what (e.g. GPS, or other info) information it made the decision. The authors needs to describe this somewhere briefly.
  4. The paper did not compare its results with any other similar work, for example, the efficacy of the TP planning in avoiding wane road or potholes. If the authors are not able to find similar work to compare with or it is not easy to repeat other's work, the author could compare with the reported results of other work. In general, the authors need to explain why no comparison is given in the paper, or add the comparison result in revision.

Authors Answer:

Ad 1: We corrected the indicated errors and noted the change in our rebuttal. We also read the paper carefully to find additional errors.

Ad 2: We corrected the figures and equations and some of their descriptions as indicated in the comment file and noted the changes in our answers above.

Ad 3:   We thank the reviewer for this comment.  In simulation, which is topic of this paper, the vehicle position is compared with the definition of the zones.  In real world tests, which will follow in the near future, the vehicle must be equipped with a GNSS receiver to determine its position.

We added a paragraph in the paper to address this concern. (Page 5, line 183)

The ego vehicle has to be equipped with a GNSS-receiver and compares its position with the definitions of the detection and relevance zones to evaluate if it is inside of a zone.

Ad 4:   We thank the reviewer for his/her remark. We agree with the reviewer that it would make a perfect sense to compare our results with any similar solution. However, we could not find any specific analysis of a trajectory planer for avoiding road damages, potholes or traveling with in-lane offsets. Therefore, we unfortunately couldn’t include any such comparison.  

Reviewer 2 Report

Review of paper 1339931

Researching, ”Development and Verification of Infrastructure-assisted Automated Driving Functions” are the ways to advance with the improvement and implementation of automated vehicles and driving. The paper presents an introduction into the subject with 18 references to the specific literature, a chapter entitled “Descriptions of the Use Case Scenario and Performance Indicators” which is defining recommended scenarios, a third chapter with simulation application / environment and simulation setup, a special chapter with automated driving functions development as a mathematical model structure, a chapter for simulation results and the sixth chapter with conclusions. The paper is thus structured on 6 chapters and has also a Acknowledgements section. The reference list consists in 21 titles. There are multiple electronic applications used for virtual modeling of the automated driving functions. The chapter two of the paper has proposed lane change recommendation scenario description and “In-lane offset recommendation scenario description”. The final conclusions and discussions are describing a new model for operation of automated driving functions, namely by using specific infrastructure routing recommendations for enhancing or adapting their behavior. Also, combined in-lane offset and lane change recommendations with varying lateral offset values are presented in figure 7. The lane boundaries were shown to aid the visualization as results of the simulations and modeling. The conclusions are complex, covering of most important aspects and laborious.

See the following:

There are some abbreviations that are not explained and defined in the paper.

The abstract should be completed with specification of paper’s objectives, results, and most important conclusions.

Please define all the terms / abbreviations used in the paper.

Author Response

Open Review

English language and style

( ) Extensive editing of English language and style required
( ) Moderate English changes required
(x) English language and style are fine/minor spell check required
( ) I don't feel qualified to judge about the English language and style

Yes

Can be improved

Must be improved

Not applicable

Does the introduction provide sufficient background and include all relevant references?

(x)

( )

( )

( )

Is the research design appropriate?

()

(x)

( )

( )

Are the methods adequately described?

()

(x)

( )

( )

Are the results clearly presented?

()

(x)

( )

( )

Are the conclusions supported by the results?

(x)

( )

( )

( )

Comments and Suggestions for Authors

Review of paper 1339931

Researching, ”Development and Verification of Infrastructure-assisted Automated Driving Functions” are the ways to advance with the improvement and implementation of automated vehicles and driving. The paper presents an introduction into the subject with 18 references to the specific literature, a chapter entitled “Descriptions of the Use Case Scenario and Performance Indicators” which is defining recommended scenarios, a third chapter with simulation application / environment and simulation setup, a special chapter with automated driving functions development as a mathematical model structure, a chapter for simulation results and the sixth chapter with conclusions. The paper is thus structured on 6 chapters and has also a Acknowledgements section. The reference list consists in 21 titles. There are multiple electronic applications used for virtual modeling of the automated driving functions. The chapter two of the paper has proposed lane change recommendation scenario description and “In-lane offset recommendation scenario description”. The final conclusions and discussions are describing a new model for operation of automated driving functions, namely by using specific infrastructure routing recommendations for enhancing or adapting their behavior. Also, combined in-lane offset and lane change recommendations with varying lateral offset values are presented in figure 7. The lane boundaries were shown to aid the visualization as results of the simulations and modeling. The conclusions are complex, covering of most important aspects and laborious.

 See the following:

There are some abbreviations that are not explained and defined in the paper.

The abstract should be completed with specification of paper’s objectives, results, and most important conclusions.

Please define all the terms / abbreviations used in the paper.

Autors’ Answer:

Thank you for your kind summary.

Ad Abbreviations: You were right, we have forgotten the explanations of some abbreviations. We hope we have now added all missing full text terms.

Ad completion of the abstract:  We revised the abstract and added some sentences, but we had to delete other parts, because of the 200 word-limit of the abstract.

Reviewer 3 Report

This paper demonstrates two ADAS use cases of an ESRIUM project in simulation based on information enabled by vehicle-to-infrastructure communication. The paper is clearly written and not difficult to follow. 

The biggest concern is the novel contribution of this work. Even though the two use cases are specially designed ADAS functions, they are very similar to normal lane changes. The underlying vehicle path planning and control strategies are then similar to many existing automated lane change functions. Thus there are limited new contributions on this aspect. The authors talked about the potential benefits of these special ADAS functions to the road surface, however, such benefits are not illustrated in the simulation. Overall, the new contribution is not clear and it is more like proposing new ADAS functions that should be included in a standard for connected and automated vehicles rather than a typical academic paper.

Author Response

Open Review

English language and style

( ) Extensive editing of English language and style required
( ) Moderate English changes required
(x) English language and style are fine/minor spell check required
( ) I don't feel qualified to judge about the English language and style

Yes

Can be improved

Must be improved

Not applicable

Does the introduction provide sufficient background and include all relevant references?

()

(x)

( )

( )

Is the research design appropriate?

()

( )

(x)

( )

Are the methods adequately described?

()

( )

(x)

( )

Are the results clearly presented?

()

( )

(x)

( )

Are the conclusions supported by the results?

()

( )

(x)

( )

Comments and Suggestions for Authors

This paper demonstrates two ADAS use cases of an ESRIUM project in simulation based on information enabled by vehicle-to-infrastructure communication. The paper is clearly written and not difficult to follow. 

The biggest concern is the novel contribution of this work. Even though the two use cases are specially designed ADAS functions, they are very similar to normal lane changes. The underlying vehicle path planning and control strategies are then similar to many existing automated lane change functions. Thus there are limited new contributions on this aspect. The authors talked about the potential benefits of these special ADAS functions to the road surface, however, such benefits are not illustrated in the simulation. Overall, the new contribution is not clear and it is more like proposing new ADAS functions that should be included in a standard for connected and automated vehicles rather than a typical academic paper.

Authors’ Answer:

We thank the reviewer for his/her positive comments and constructive criticism.

About the “Open Review Comments” we are happy that the reviewer found the paper clear despite few grammatical errors. We have tried to review/go through the whole paper and correct all small grammatical mistakes and typos in the current revision. On the other hand, we are sorry read that the Reviewer evaluated that the paper needs or can be improved in almost all the fields. We have tried to conduct all the recommended specific changes in the specific reviewer comments. We now sincerely hope that our revised version will be sufficient to improve this evaluation.

We thank the reviewer for the specific comment

    “This paper demonstrates two ADAS use cases of an ESRIUM project in simulation based on information enabled by vehicle-to-infrastructure communication. The paper is clearly written and not difficult to follow.”

We further would like to express that it is not sole aim of our paper to demonstrate the use cases of ESRIUM. It is rather the specific toolchain/software and the algorithms to realize these use cases in real life. This paper represents the first step in this direction by implementing the complete solution in simulation. In the next step these algorithms and the exact algorithms will be tested in real time on a real motorway with real infrastructure recommendations. At this stage the simulation results will represent a basis for comparison of the developed solution.

"The biggest concern is the novel contribution of this work. Even though the two use cases are specially designed ADAS functions, they are very similar to normal lane changes. The underlying vehicle path planning and control strategies are then similar to many existing automated lane change functions. Thus there are limited new contributions on this aspect. The authors talked about the potential benefits of these special ADAS functions to the road surface, however, such benefits are not illustrated in the simulation. Overall, the new contribution is not clear and it is more like proposing new ADAS functions that should be included in a standard for connected and automated vehicles rather than a typical academic paper."

We are sorry that the Reviewer thinks that the contribution of the paper is not clear. As written above, the solution developed in this work does not exist and is not realizable by any driving function available at present. There are off course similarities with existing driving functions and the specific recommendations are look-like simple double lane change maneuvers. However, the overall similarity ends here. The main difference is that these lane changes and lane offsets are conducted to avoid the road damage locations and ruts so that to prevent any further damage to the vehicle and the road. The automated generation of a  “Map Layer” to detect the road wear which is to be used for generating the routing recommendations is also studied in the scope of the ESRIUM project. We would very much happy if our studies in ESRIUM can contribute to standardization of such routing recommendations in the future. We believe that we are the very first ones to investigate this problem and above all the first ones to demonstrate in motorway driving condition on the public roads (which are stated in the paper as future work already)